# Tumor Heterogeneity: A Great Barrier in the Age of Cancer Immunotherapy

**DOI:** 10.3390/cancers13040806

**Published:** 2021-02-15

**Authors:** Nader El-Sayes, Alyssa Vito, Karen Mossman

**Affiliations:** 1Department of Biochemistry and Biomedical Sciences, McMaster Immunology Research Centre, McMaster University, Hamilton, ON L8S 4K1, Canada; elsayesn@mcmaster.ca (N.E.-S.); vitoar@mcmaster.ca (A.V.); 2Department of Medicine, McMaster Immunology Research Centre, McMaster University, Hamilton, ON L8S 4K1, Canada

**Keywords:** tumor heterogeneity, antigen escape, immunotherapy, cancer

## Abstract

**Simple Summary:**

Despite great advances in cancer therapy, tumor heterogeneity continues to be a great barrier for the successful treatment of cancer. It has long been established that tumor heterogeneity is prevelant in most cancer patients and is a major driver of acquired resistance to all forms of cancer therapy. In the case of immunotherapy, the response of the immune system against specific tumor antigens can drive selective pressure towards antigen-negative cells, which is a common cause of relapse in the clinic. In this review we summarize the mechanisms in which tumor heterogeity can arise. Furthermore, we discuss recent advances in the field of oncology that can be used to better identify, study, and overcome tumor heterogeneity.

**Abstract:**

Throughout the history of oncology research, tumor heterogeneity has been a major hurdle for the successful treatment of cancer. As a result of aberrant changes in the tumor microenvironment such as high mutational burden, hypoxic conditions and abnormal vasculature, several malignant subpopulations often exist within a single tumor mass. Therapeutic intervention can also increase selective pressure towards subpopulations with acquired resistance. This phenomenon is often the cause of relapse in previously responsive patients, drastically changing the expected outcome of therapy. In the case of cancer immunotherapy, tumor heterogeneity is a substantial barrier as acquired resistance often takes the form of antigen escape and immunosuppression. In an effort to combat intrinsic resistance mechanisms, therapies are often combined as a multi-pronged approach to target multiple pathways simultaneously. These multi-therapy regimens have long been a mainstay of clinical oncology with chemotherapy cocktails but are more recently being investigated in the emerging landscape of immunotherapy. Furthermore, as high throughput technology becomes more affordable and accessible, researchers continue to deepen their understanding of the factors that influence tumor heterogeneity and shape the TME over the course of treatment regimens. In this review, we will investigate the factors that give rise to tumor heterogeneity and the impact it has on the field of immunotherapy. We will discuss how tumor heterogeneity causes resistance to various treatments and review the strategies currently being employed to overcome this challenging clinical hurdle. Finally, we will outline areas of research that should be prioritized to gain a better understanding of tumor heterogeneity and develop appropriate solutions.

## 1. Introduction

Major challenges with universal cancer therapy have historically been attributed to the large number of subtypes of the disease and the biological differences associated with cancers arising in different parts of the body. While this locational diversity remains a challenge for unifying cancer treatment across various types, it has now become clear that even patients with phenotypically identical cancers often have dichotomous responses to treatment [1,2]. As we have continued to investigate the inner workings of a tumor, we have discovered that cancers are formed by many distinct cellular populations, rather than a homogenous cluster of identical cells. This tumor heterogeneity can take the form of cellular and genetic heterogeneity. Cellular heterogeneity is used to describe the diversity of cells that can be found in the tumor microenvironment (TME). The presence of cancer-associated fibroblasts, endothelial cells and immune cells plays an important role in tumor progression and response to therapy. In this review, however, we will focus on the genetic heterogeneity of cancer cells within one or several tumors, which is often a driver of acquired resistance in tumors and represents a major hurdle for proper diagnosis, prognostic prediction and therapeutic efficacy [3]. 

It is well established that tumor heterogeneity is largely driven by aberrant changes in the TME, such as high mutational burden, hypoxic conditions and abnormal vasculature [4]. Additionally, native TME factors aside, the administration of therapy can result in selective pressure towards subpopulations with acquired resistance mechanisms [3,5]. This phenomenon is often the cause of relapse in previously responsive patients, drastically shifting expected therapeutic outcomes. As oncology research has progressed and evolved, so too has our understanding of cancer biology and the immunological synapse involved in swaying clinical prognoses. But the question still remains, how do these environmental and therapeutic situations influencing the tumor create heterogeneity and distinctive cellular populations? 

In 1976 Peter Nowell published a landmark perspective piece suggesting that cancer arises through an evolutionary process, similar in nature to Darwinian evolution and natural selection [6]. Nowell suggested that cancers arise through stepwise, somatic cellular mutations inevitably leading to multiple sub-clonal populations (Figure 1). This revelation changed the way that cancer was studied and further highlighted the need for multi-pronged therapeutic approaches and a deeper understanding of cancer at a genomic level. Indeed, as therapeutic approaches have evolved in recent decades, we have seen an insurgence of combined therapeutic modalities that can target cancer cells through differing mechanisms, with the goal of overcoming innate acquired resistance.

In the last decade it has become evident that identifying the biological drivers of cancer will ultimately lead to personalized cancer treatment and improved clinical outcomes [7]. As such, oncologists now increasingly use molecular characterization of a sample from a primary or metastatic tumour lesion to guide their selection of treatments for an individual patient. However, this approach can prove problematic as they rely on a limited sample of cancer tissue, often obtained from a needle biopsy or surgical excision, that is unlikely to accurately capture the complete genomic landscape of a patient’s cancer. For example, clinical instances have occurred in which estrogen receptor (ER) expression in a primary breast cancer sample does not mirror what is later found in a distant metastatic lesion. In fact, in 7–25% of patients’ ER expression was different between the two sites [8,9,10]. Changes in ER status may have important clinical implications because patients with tumors that lack ER expression do not benefit from treatment with endocrine therapy such as tamoxifen or aromatase inhibitors [11]. This is just one example of how temporal heterogeneity can affect clinical outcomes. Furthermore, comprehensive characterization of multiple tumor specimens obtained from the same patient illustrates that remarkable intratumour heterogeneity can exist not only in the temporal sense as in the breast cancer example, but also between geographical regions in the same tumour.

Cancer genomics studies, including large-scale collaborative sequencing projects such as The Cancer Genome Atlas (TCGA) and the International Cancer Genome Consortium (ICGC), have catalogued genetic interpatient tumour heterogeneity for cancers of the same histological subtype. Non-genetic phenotypic and functional heterogeneity is also well recognized, as is heterogeneity within the TME. As high throughput technologies become more affordable and readily accessible, researchers continue to expand their understanding of the factors that influence tumor heterogeneity and shape the TME not only at the time of diagnosis, but also over the course of treatment regimens. 

Here, we review the clinical implications of tumour heterogeneity for cancer diagnosis, prognostic predictions, treatment selection and resistance. We will investigate the factors that give rise to tumor heterogeneity and the impact it has on both established and emerging therapeutic regimens. We will discuss how tumor heterogeneity causes resistance to various treatments and review the strategies currently being employed to overcome this challenging clinical hurdle. Additionally, we will discuss how clinical trials that are restricted to molecular subtypes of cancer could incorporate studies of tumour heterogeneity so that we can better understand the clinical impact of heterogeneity on therapeutic efficacy and the emergence of acquired resistance. 

## 2. Factors Contributing to Tumor Heterogeneity

Tumor heterogeneity can take different forms, each posing a unique challenge for successful treatment of disease and for overcoming the risk of acquired resistance (Figure 2). Patients with the same type of malignancy may experience vastly different clinical outcomes, both before and after treatment. This interpatient heterogeneity is often seen in the clinic and is largely attributed to differences in somatic mutations acquired in the tumor. Patients acquire mutations not only in different genes, but also in different domains within the same gene [12]. Heterogeneity is also prevalent within an individual patient and can take the form of intratumor, intermetastatic or intrametastatic heterogeneity. Intratumor heterogeneity consists of a single tumor mass which contains several distinct subpopulations of cells, each behaving differently with varied responses to therapeutic intervention [13]. Furthermore, intermetastatic heterogeneity can be observed between different tumor masses, as subpopulations of cells can migrate from the primary tumor and travel to distant sites within the body to form metastatic lesions. In fact, heterogeneity between different malignant lesions is the most common clinical observation in patients with advanced metastatic disease [13,14,15]. Like intratumor heterogeneity, intrametastatic heterogeneity is defined by multiple cellular subpopulations within a single metastatic lesion. In the past decade there has been an increase in research focused on studying tumor heterogeneity and its role in developing acquired resistance to many different types of cancer therapies. While assessing the emergence of heterogenous cell populations throughout the course of tumorigenesis, there has been some debate as to whether heterogenous populations are pre-existing in the tumor or developed in response to therapeutic intervention. In reality, many factors may lead to heterogeneity during tumor progression, however some types of therapy also contribute to genomic diversity by increasing the tumor mutational burden.

There have been many studies showing that heterogenous populations arise during the early stages of tumor progression, and that treatment leads to selective pressure towards resistant populations [16,17,18,19,20]. Among the many factors that drive tumor heterogeneity, genomic instability is most prominent in all malignancies. Many of the biological hallmarks associated with cancer development, such as limitless replicative potential, increase the mutational rate and genomic instability of malignant cells, which in turn give rise to other malignant traits [21]. This cascading effect often results in heterogeneity in the tumor as different cells acquire unique mutations that give rise to genetically distinct subpopulations [5,6,22]. For example, in normal cells the tumor suppressor p53 responds to DNA damage by triggering pathways involved in cell-cycle arrest, apoptosis, or DNA repair [23]. Mutations in p53 are some of the most frequent mutations found in human cancers and are largely associated with tumorigenesis and genomic instability [24,25]. Other hallmark traits associated with tumorigenesis may further contribute to genomic instability. For example, hypoxic conditions are often found in the TME due to the enhanced cellular kinetics and increased replicative potential that require greater oxygen uptake [4]. In one study, Ma and colleagues sampled the single-cell transcriptomic landscape of 19 patients with liver cancer [26]. They found that hypoxia-dependant VEGF expression was associated with higher transcriptomic diversity in the tumor. Furthermore, tumors with more heterogeneity contained T cells with lower cytotoxic activity and were associated with worse overall survival [5]. Indeed, hypoxia is well known to impede mismatch repair, cause genomic instability and promote the formation of subclones in the TME [4,27,28]. 

While tumor heterogeneity is unavoidable during tumor progression, some therapeutic interventions can further drive the formation of genetic and epigenetic diversity in the tumor. Many chemotherapies are DNA-damaging agents or agents that interfere with DNA replication/repair pathways [29,30]. This type of cellular damage results in the risk that persisting cancer cells may develop greater levels of genomic instability that can give rise to resistant populations [5]. Indeed, a study lead by McGranahan et al. found an association between chemotherapy and neoantigen heterogeneity. Furthermore, neoantigen heterogeneity was associated with poor clinical outcomes in response to PD-1 and CTLA-4 blockade [31]. This concern also extends to radiation therapy, as ionizing radiation can also contribute to genomic instability and carcinogenesis [32]. 

## 3. Acquired Resistance and Antigen Escape

There is a plethora of innovative therapeutic approaches currently being developed for the treatment of cancer. Tumor heterogeneity acts as a major hurdle for treatment and a potentiator of acquired resistance regardless of the therapeutic approach or the type of cancer. Unlike primary resistance, acquired resistance can occur in patients that initially respond to therapy, resulting in a relapse after a period of tumor regression [33]. The mechanism of resistance caused by tumor heterogeneity is the same regardless of the treatment received. In a manner similar to that of natural selection, the composition of subpopulations in the tumor changes dynamically as a result of selective pressure exerted by therapeutic intervention and changes in the TME. Indeed, relapse of chemoresistant tumors is frequently observed in the clinic [5,34,35]. One such example was highlighted in a study led by Kim et al. in which 20 patients with triple negative breast cancer (TNBC) were observed during neoadjuvant chemotherapy (NAC). They identified 10 patients with persistent chemoresistant clones after treatment. Upon further analysis using single-cell DNA and RNA sequencing, the data indicated that the resistant clones were pre-existing and adaptively selected by NAC [20]. A similar study involved genetic and histological analyses of tumor biopsies from 37 patients with non-small cell lung cancer (NSCLCs) that developed resistance to treatment with EGFR inhibitors. Many of these tumors acquired various mechanisms of resistance, including mutations in the PIK3CA gene and epithelial to mesenchymal transition. Furthermore, some of the tumors lost their mechanisms of resistance in the absence of further EGFR inhibitor treatment and were later responsive to a secondary challenge with EGFR inhibitors [36]. 

Cancer immunotherapy has made extraordinary strides in the past decade, with the emergence of a variety of antibody, cell-based and virus-based therapies becoming first line treatments for some cancers. Unfortunately, populations with acquired resistance to immunotherapy can also arise because of tumor heterogeneity. The immune system can both hinder and promote tumor progression, a process known as cancer immunoediting. Cancer immunoediting can proceed through 3 distinct stages termed elimination, equilibrium and escape [37,38]. During the elimination phase, innate and adaptive immune cells work together to identify and eliminate malignant cells early during tumor progression. Some cancer cell variants may evade elimination and enter a phase of equilibrium, in which the immune system prevents further tumor progression but can inadvertently modify the immunogenicity of the cancer cells. Finally, the genetic instability of the cancer cells combined with the constant selective pressure of the immune system can give rise to new cancer cell variants that escape immune control. In the case of immunotherapy, response against specific antigens exerts selective pressure towards antigen-negative subclones over time, a concept termed antigen escape (Figure 3) [39]. An example of antigen escape was noted in two patients with stage IV melanoma treated by adoptive T-cell transfer. Expression of the T-cell-recognized neoantigen was lost in these patients, although it is unclear whether this form of antigen loss was caused by downregulation of the target antigen or the selection of antigen-negative variants [40]. In another study conducted by Sotillo and colleagues, samples were obtained from patients with B-cell acute lymphoblastic leukemias (B-ALL) that relapsed from CD19-specific CAR T-cell therapy. They found that relapsed cells contained alternatively spliced *CD19* mRNA with an omission of exon 2. The alternatively spliced form results in an N-terminally truncated form of CD19 that retains some functionality but avoids CAR T-cell-mediated killing [41]. 

As previously discussed, antigen escape has become a well-documented hurdle for immunotherapy platforms that are designed to target specific antigens. At first glance this may indicate that spontaneous antitumor immunity is favorable over targeted responses and is less susceptible to acquired resistance mediated by antigen escape. Immunotherapy platforms such as immune checkpoint blockade (ICB) and in situ vaccines are examples of therapies that can potentiate spontaneous antitumor responses, however spontaneous CD8^+^ T-cell immunity will likely be restricted to a few dominant neoantigens [42,43,44,45]. Therefore, spontaneous antitumor responses can also exert selective pressure towards antigen-negative clones. In one study, mice bearing B16 murine melanoma were vaccinated with plasmids encoding hsp70 and the HSVtk suicide gene. The authors found that suboptimal plasmid vaccination selected for aggressive tumor variants that lost the immunodominant neoantigen while retaining the expression of other known melanoma antigens [46]. Similar conclusions can be made from another study that examined the evolving landscape of tumor neoantigens during the emergence of acquired resistance in patients with NSCLC after treatment with ICB therapy. They report that resistant tumors lost the expression of neoantigens that were recognized by the host antitumor T-cells [47]. Shifts from dominant to secondary neoantigens have been observed in some cases and will be discussed further in Section 5, however shifts in antigen specificity are rare occurrences in the absence of deliberate intervention. The prevalence of antigen escape as a mechanism of acquired resistance to immunotherapy underscores the importance of further research to understand and manipulate the dynamic landscape of the TME.

## 4. Strategies for Identifying Tumor Heterogeneity

Since the first report of cancer genome sequencing appeared in 2006 [48], research involving genomic data has gained traction, with clinical implications rapidly revealing themselves. Initially, progress in obtaining such large datasets was hindered by the high cost and limited availability of whole tumor sequencing, preventing researchers and clinicians from readily taking advantage of the technology. However, over the years sequencing facilities have become more widespread and costs have gradually reduced, allowing for an explosion of cancer genomic data and publicly available datasets. 

In 2006 the TCGA program began, a landmark cancer genomics program that characterized over 20,000 primary cancer and matched normal samples spanning 33 cancer types. This joint effort between the National Cancer Institute and the National Human Genome Research Institute brought together researchers from a wide range of disciplines and multiple institutions. Over a span of 12 years, TCGA generated over 2.5 petabytes of genomic, epigenomic, transcriptomic and proteomic data. These datasets were released publicly and have since been used to improve diagnosis, treatment, and cancer prevention with over 11,000 publications utilizing the data from this massive collaborative effort. 

The goal of sequencing cancer genomes is to develop a better understanding of factors that potentiate or hinder tumor progression at cellular and molecular levels. Sequencing data can be used for biomarker discovery to improve diagnoses, drug target discovery for therapeutic intervention and personalized medicine by matching patients with the treatment most likely to be efficacious against their disease. In the case of heterogeneity, the latter is of particular importance as sequencing efforts can help to identify heterogeneity within a tumor and also within multiple lesions from the same patient. However, most datasets obtained have one major limitation: clinical details of the sample donors are often incomplete or missing altogether. In fact, the first cohort of samples collected for TCGA were complimented with only the donor’s gender, diagnosis and age at diagnosis. Key clinical data such as the administered therapy and clinical outcomes were often not included. The future of cancer genome sequencing aims to eliminate these restraints but will have to overcome strict patient privacy laws and the lack of a centralized hub for all data. 

In the context of heterogeneity, the more recent development of single-cell RNA sequencing (scRNA-seq) is particularly exciting, as it enables the determination of not only the frequency of individual mutations, but also determination of co-occurring and mutually exclusive alterations. Indeed, scRNA-seq has been used to demonstrate the presence of multiple cell populations within a tumor, each belonging to a distinct molecular group [49,50,51]. Furthermore, Kinker and colleagues have used scRNA-seq to investigate the ability of cultured cell lines to recapitulate the heterogeneity observed among malignant cells in human tumors [52]. They profiled 198 cancer cell lines from 22 cancer types and identified 12 expression programs as being heterogenous across multiple cancer cell lines. These programs, including cell cycle, senescence, stress and interferon responses, were further shown to recapitulate those recently identified as being heterogenous in expression within human tumors. This information allowed the researchers to identify specific cell lines as being the most relevant models of cellular heterogeneity, which they then used to study subpopulations of senescence-related cells, demonstrating their unique drug sensitivities, which were predictive of clinical response. This extensive and thorough body of work is a prime example of how scRNA-seq can be used to identify recurrent patterns of heterogeneity that are shared between tumors and the models we use for preclinical development. 

Circulating tumor DNA (ctDNA), released from both normal and cancerous cells, has recently arisen as an exciting new biomarker in the field of oncology. ctDNA is characterized as DNA that contain genetic changes that can be used for identifying cancer. Many studies have reported impressive clinical data with cancer detection accuracy ranging from 50–70% [53,54,55]. Genomic characterization of ctDNA or circulating tumour cells may offer an opportunity to assess clonal dynamics throughout the course of a patient’s illness and identify drivers of therapeutic resistance. In one study, Ma and colleagues showed that ctDNA can be used to assess tumor heterogeneity and predict treatment outcomes in metastatic breast cancer [56]. 

High throughput technologies such as scRNA-seq and innovative biomarker discovery strategies such as ctDNA are just some of the ways that the field of cancer research is rising to the challenge of identifying and tackling tumor heterogeneity. These technologies are becoming cheaper and more accessible, enabling a deeper understanding of the frequent changes in the TME and for the monitoring in both pre-clinical and clinical settings. These developments allow for more personalized solutions to the ever-changing landscape of tumors and can be used to predict the efficacy of therapeutic combination strategies, improving clinical outcomes and sparing patients from aggressive therapies that may provide no benefit to their tumor phenotype. 

## 5. Emerging Strategies to Overcome Tumor Heterogeneity

The strong association between tumor heterogeneity and poor clinical prognosis has rekindled research for developing strategies to overcome tumor heterogeneity (Table 1). The oldest and most common approach is to combine therapies with different mechanisms of action in a multi-pronged attempt to prevent the selection of resistant populations. Combination therapies have become commonplace for classical chemotherapy, with most therapeutic regimens involving the combination of different chemotherapeutic agents with or without surgical resection [57]. This strategy has extended to combining chemotherapy and immunotherapy with promising pre-clinical and clinical results. One meta-analysis of 12 phase-III clinical trials highlights the benefits of combining chemotherapy with atezolizumab and/or pembrolizumab in patients with NSCLC [58]. A similar example can be seen in a phase-III clinical trial in which the combination of atezolizumab and nab-paclitaxel showed improved progression-free survival in patients with metastatic TNBC [59]. In a pre-clinical study by Nguyen et al., the authors utilized a combination strategy to eliminate antigen-negative tumors that arose after treatment with adoptive T cell therapy (ACT) followed by oncolytic virus vaccination. They found that the addition of a class I histone deacetylase inhibitor (HDACi) to the therapeutic regimen reprogrammed immunosuppressive tumor-infiltrating myeloid cells to eliminate antigen-negative tumor cells [60]. Indeed, many studies have demonstrated the efficacy of chemotherapy when combined with ICB [61,62], oncolytic viruses [63,64,65], and cell-based immunotherapies [66,67,68]. As discussed previously, some therapies may contribute to the genomic instability in cancer cells that can give rise to resistant subclones. Instead of attempting to prevent more genomic instability in the tumor, some combination therapies are designed to exploit this property for therapeutic benefit. Zhang and colleagues report that cyclin-dependent kinase 7 (CDK7) can be used to potentiate genomic instability, which triggers an antitumor immune response. Addition of anti-PD-1 therapy enhanced the antitumor response and offered a significant survival benefit in a murine small cell lung cancer model [69]. 

Combination therapies for the treatment of cancer often outperform their respective monotherapies in progression-free survival and overall survival, however this does not guarantee the prevention of resistant subclones relapsing after initial treatment. In the context of immunotherapy, antigen escape is often a source of acquired resistance. One method to prevent antigen escape is to expand the antitumor response to generate a more diverse immune repertoire that would be better suited to target heterogenous tumors. Data from several studies suggest that multi-peptide vaccines can be designed to create a response against several tumor antigens simultaneously [70,71,72]. In one of these studies, the authors designed a multi-antigen vaccine for renal cell cancer patients that increases the breadth of the immune response and resulted in better disease control compared to patients that responded to a single peptide [72]. Perhaps one of the greatest drawbacks to a multi-antigen vaccine strategy is the lack of known tumor antigens in many cancers. However, new generation high throughput technologies have enabled great strides for identifying tumor neoantigens [44]. In two preclinical studies, an in silico approach was used to identify immunogenic neoantigens which were then used to design multi-antigen vaccines [73,74]. Ideally, incorporating in silico and high throughput approaches could be used to design multi-antigen vaccines throughout the dynamic course of treatment and tumor progression. 

Finally, enhancing the T-cell repertoire through antigen spread could allow for a broadened immune response regardless of the primary response. Antigen spread (also known as determinant spread and epitope spread) is the expansion of an immune response from a dominant antigen to secondary antigens. Expansion to secondary antigens could include different epitopes from the same antigen or from other antigens [75]. Antigen spread can occur during cell lysis, which releases potential secondary antigens that can be taken up and cross-presented by antigen presenting cells [75]. Interestingly, antigen spread has been observed in the clinic and is associated with improved prognosis [76]. In one clinical study, patients with stage III–IV melanoma were treated with autologous dendritic cells pulsed with an immunodominant epitope (MART-1 27-35). While MART-1-specific immunity did not correlate with clinical outcome, the only patient with a complete response developed immunity against other melanoma epitopes that were not included in the vaccine [77]. Antigen spread has been observed in several other pre-clinical and clinical studies [78,79,80,81], however it is unclear if antigen spread can be deliberately induced through therapeutic intervention. One group demonstrated that radiation therapy can cause an expansion of the T-cell repertoire in a murine melanoma model, which allowed for improved survival when combined with ICB [82]. As previously mentioned, antigen spread likely occurs because of cell lysis in the TME, which suggests that immunogenic forms of cell death could be used to potentiate antigen spread, however more research is needed to support this theory. 

**Table 1 cancers-13-00806-t001:** Therapeutic strategies to combat tumor heterogeneity. ICB = Immune checkpoint blockade, OV = oncolytic virus, ACT = adoptive T cell transfer, OVV = oncolytic virus vaccine, HDACi = class I histone deacetylase inhibitor, CDK7 = cyclin-dependent kinase 7, RCC = renal cell carcinoma.

Approach	Rationale	Examples
Combination Therapies	Using a multi-pronged approach to target multiple pathways simultaneously, preventing selection of resistant populations	1. Chemotherapy + ICB improves progression-free survival in patients [58,59,61,62] 2. Chemotherapy + OV improved therapeutic efficacy [63,64,65] 3. Chemotherapy + cell-based therapies [66,67,68] 4. ACT + OVV + HDACi reprogrammed immunosuppressive myeloid cells, eliminating antigen-negative tumor cells in mice [60] 5. CDK7 + ICB enhanced antitumor immunity and prolonged survival outcomes in mice [69]
Multi-Peptide Vaccines	Designed to create a response against several tumor antigens simultaneously	1. Multi-antigen vaccine in RCC patients increased the breadth of the immune response and resulted in better disease control [72] 2. Immunogenic neoantigens were first identified and then used to design multi-antigen vaccines, improving therapeutic outcomes [73,74]
Antigen Spread	Intended to enhance the T-cell repertoire, allowing for expansion of the immune response from a dominant antigen to secondary antigens	1. Autologous DCs pulsed with an immunodominant epitope resulted in antigen spread in one patient, resulting in a complete response to treatment [77] 2. Radiation therapy can expand T-cell repertoire, allowing for improved survival when combined with ICB [82]

## 6. Conclusions and Future Perspectives

The phenomenon of tumor heterogeneity and clonal evolution in cancers has long been identified as a major driver of tumor development, metastasis and acquired resistance mechanisms. The availability of next-generation sequencing and advances in the field of bioinformatics have enabled clinicians to assess this previously elusive phenomenon in real-time clinical settings. However, as we uncover new ways of detecting and monitoring heterogeneity, we must also in parallel assess the ways that various forms of heterogeneity contribute to clinical outcomes across the full spectrum of cancer types and therapeutic modalities. 

In order to fully understand the natural progression of tumor heterogeneity and the clinical implications associated with various forms of therapy, it is important that clinical trial design incorporates ways of assessing heterogeneity into newly developed studies. For high level reproducibility in clinical research and diagnostics, it will be necessary to establish streamlined, standardized analytical methods. For example, liquid biopsies for plasma DNA analysis are currently recognized as the best method to study recurrent tumors that are refractory to therapy, and widespread clinical use of this technique could be highly beneficial to the field. Indeed, it has already been suggested that at least two types of methodological approaches should be considered to assess clinical heterogeneity [83]. Stanta and colleagues have reasonably suggested that surgically treated tumors should undergo a thorough analysis of tissues to drive the appropriately selected adjuvant therapy, and in recurrent cancer, follow-up should consider the inclusion of blood analysis of ctDNA. Another example is the use of radiomics, which is an emerging field of medical image analysis that utilizes radiological images to predict patient outcomes. It has been proposed that radiomics could be used to quantify tumor heterogeneity [84,85]. Since radiological images are frequently taken for diagnostic purposes, the ability to track tumor heterogeneity throughout tumor progression would be extremely valuable for furthering our understanding of tumor heterogeneity. Careful consideration of inclusion of such techniques into evolving clinical practice will further increase the wealth of data available to researchers and clinicians in an effort to identify correlative changes. 

As the field of immunotherapy continues to progress and immunotherapies solidify their place as a true pillar of cancer therapy, the clinical hurdle of tumor heterogeneity so too moves to the forefront of oncologic research. As we study the immunological synapse associated with immunotherapy outcomes and in particular, patients who are refractory to treatment with immunotherapy, we need to simultaneously assess the dynamic shifts of the TME and heterogeneity levels. Incorporating innovative, high-level techniques, we can move cancer treatment into the realm of personalized medicine and monitor patients and their response to therapy in real-time clinical settings. Understanding the evolutionary drivers for heterogeneity will be key in mapping out primary tumorigenesis, metastatic formation and relapsed disease as we continue to work towards improving outcomes and quality of life for patients affected by cancer. 

## Figures and Tables

**Figure 1 cancers-13-00806-f001:**
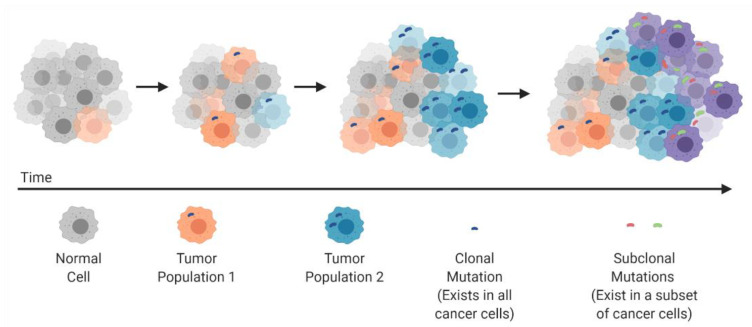
Clonal evolution and development of tumor heterogeneity.

**Figure 2 cancers-13-00806-f002:**
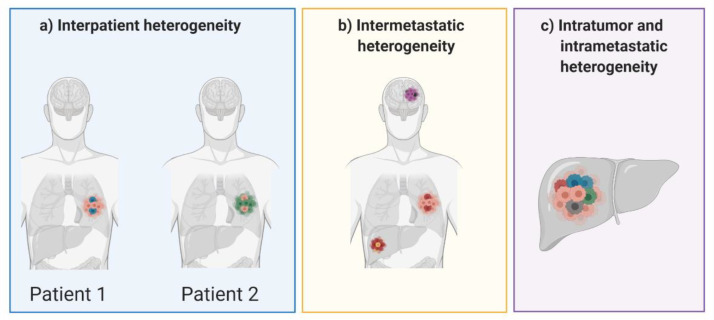
Forms of tumor heterogeneity that can occur between patients and in individual patients.

**Figure 3 cancers-13-00806-f003:**
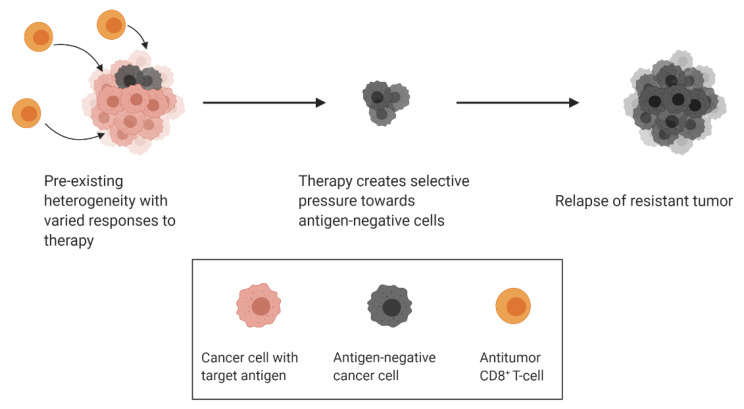
Selective pressure from antitumour T-cells drives resurgence of antigen-negative clones.

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
