# Peer review of "Tumor Heterogeneity: A Great Barrier in the Age of Cancer Immunotherapy"

_cancers, 2021, doi:10.3390/cancers13040806_

Round 1
Reviewer 1 Report
This is a comprehensive review for tumor heterogeneity. The authors provided the most recent advances in the factors, modes, and mechanisms of tumor heterogeneity and proposed the strategies for identifying and overcoming the barrier. The manuscript was well-written and the argument seems logical.
However, Fig. 1 should be revised so that the marks of submucosal mutations can be clearly visible.
In addition, it would improve the paper if they show an original figure explaining the strategies to overcome tumor heterogeneity.
A paper by Lichun Ma, et al. is missing in the references. (Page 4, Line 146)
Author Response
Reviewer 1:
This is a comprehensive review for tumor heterogeneity. The authors provided the most recent advances in the factors, modes, and mechanisms of tumor heterogeneity and proposed the strategies for identifying and overcoming the barrier. The manuscript was well-written and the argument seems logical.
However, Fig. 1 should be revised so that the marks of submucosal mutations can be clearly visible.
Author’s response: We thank the reviewer for their positive feedback and helpful suggestions. We adjusted the colors of the mutations in figure 1 to make them more visible against the background (page 2).
In addition, it would improve the paper if they show an original figure explaining the strategies to overcome tumor heterogeneity.
Author’s response: We thank the reviewer for this suggestion; however, we believe that the included table is more effective for summarizing the existing literature while also providing quick access to the relevant references. We feel that the addition of a figure might be redundant and would not add significant information for the reader.
A paper by Lichun Ma, et al. is missing in the references. (Page 4, Line 146)
Author’s response: We thank the reviewer for identifying this mistake. We added the appropriate reference (page 4, line 152).
Reviewer 2 Report
The manuscript reviews an important issue of tumor heterogeneity and the impact on cancer immunotherapy. It is well written and addresses the major issues in a concise form, and specifically the effects of therapeutic interventions on the tumor heterogeneity. As such, it is valuable contribution to the field and is of interest to the readers of Cancers. However, several points need to be modified in order to improve the manuscript.
First, the authors should clearly state in the abstract and the manuscript that the topic is genetic heterogeneity and not cellular heterogeneity. Along this line, a brief description of the cellular heterogeneity of the TME (i.e. immune cells, cancer-associated fibroblasts, endothelial cells) should be included.
Second, the authors should include the concept of immunoediting on page 3, Acquired resistance and antigen escape. E.g. see landmark paper PMID: 21436444.
And third, on page 9, Future perspectives, it would be helpful to mention radiomics as a way to monitor heterogeneity.
Author Response
Reviewer 2:
The manuscript reviews an important issue of tumor heterogeneity and the impact on cancer immunotherapy. It is well written and addresses the major issues in a concise form, and specifically the effects of therapeutic interventions on the tumor heterogeneity. As such, it is valuable contribution to the field and is of interest to the readers of Cancers. However, several points need to be modified in order to improve the manuscript. First, the authors should clearly state in the abstract and the manuscript that the topic is genetic heterogeneity and not cellular heterogeneity. Along this line, a brief description of the cellular heterogeneity of the TME (i.e. immune cells, cancer-associated fibroblasts, endothelial cells) should be included.
Author’s response: We thank this reviewer for their thorough comments and suggestions in helping to improve this manuscript. Lines 21-23 in the abstract discuss specifically a diversity of malignant subpopulations in the tumor. We added a description of both cellular and genetic heterogeneity on page 2 (lines 48-53). We also added a statement to make it clear to the reader that the focus of this review will be on genetic heterogeneity.
Second, the authors should include the concept of immunoediting on page 3, Acquired resistance and antigen escape. E.g. see landmark paper PMID: 21436444
Author’s response: We thank this reviewer for their suggestion. We believe it is a valuable addition that is highly relevant to the topic. We added a description of the concept of immunoediting to section 3 (page 5, lines 193-202).
And third, on page 9, Future perspectives, it would be helpful to mention radiomics as a way to monitor heterogeneity.
Author’s response: We added a section explaining radiomics and the potential for using radiomics to identify and monitor tumor heterogeneity on page 10 (lines 393-398).
Reviewer 3 Report
This review article is well written and informative coverage of the fields of tumor heterogeneity and immunotherapy. As a reviewer, I do not find outstanding issues.
Author Response
Reviewer 3:
This review article is well written and informative coverage of the fields of tumor heterogeneity and immunotherapy. As a reviewer, I do not find outstanding issues.
Author’s response: We thank the reviewer for their positive feedback and for recommending our manuscript for publication.